# On-chip phonon-enhanced IR near-field detection of molecular vibrations

Andrei Bylinkin [1,2], Sebastián Castilla [3], Tetiana M. Slipchenko [4,5], Kateryna Domina [6], Francesco Calavalle[1], Varun-Varma Pusapati [3], Marta Autore[1], Fèlix Casanova [1,7], Luis E. Hueso [1,7], Luis Martín-Moreno [4,5], Alexey Y. Nikitin [2,7], Frank H. L. Koppens [3,8] & Rainer Hillenbrand [7,9] ✉

Phonon polaritons – quasiparticles formed by strong coupling of infrared (IR) light with lattice vibrations in polar materials – can be utilized for surface-enhanced infrared absorption (SEIRA) spectroscopy and even for vibrational strong coupling with nanoscale amounts of molecules. Here, we introduce and demonstrate a compact on-chip phononic SEIRA spectroscopy platform, which is based on an h-BN/graphene/h-BN heterostructure on top of a metal split-gate creating a p-n junction in graphene. The metal split-gate concentrates the incident light and launches hyperbolic phonon polaritons (HPhPs) in the heterostructure, which serves simultaneously as SEIRA substrate and room-temperature infrared detector. When thin organic layers are deposited directly on top of the heterostructure, we observe a photocurrent encoding the layer's molecular vibrational fingerprint, which is strongly enhanced compared to that observed in standard far-field absorption spectroscopy. A detailed theoretical analysis supports our results, further predicting an additional sensitivity enhancement as the molecular layers approach deep subwavelength scales. Future on-chip integration of infrared light sources such as quantum cascade lasers or even electrical generation of the HPhPs could lead to fully on-chip phononic SEIRA sensors for molecular and gas sensing.

Mid-infrared (mid-IR) spectroscopy is a versatile analytical tool for label-free and non-destructive identification of materials via their infrared-vibrational fingerprint[1]. However, the small infrared extinction cross-sections challenge the detection of minute amounts or concentrations of molecules. To overcome this limitation, surface-enhanced infrared absorption (SEIRA) spectroscopy has been developed[2–4]. In SEIRA spectroscopy, the sensitivity is improved by placing molecules on substrates designed to enhance the incident

infrared field, which is often based on rough metal surfaces[5–7] or metallic nanostructures[8–12] exhibiting surface plasmon polariton resonances. Further improvement can be achieved by exploiting the extraordinary infrared field confinement of plasmon polaritons in graphene[13–16]. However, plasmonic resonances typically exhibit low-quality factors, which limits field enhancements and makes it challenging to achieve vibrational strong coupling for further boosting sensitivity[4]. Higher quality factors can be achieved with dielectric

[1]CIC nanoGUNE BRTA, 20018 Donostia-San Sebastián, Spain. [2]Donostia International Physics Center (DIPC), 20018 Donostia-San Sebastián, Spain. [3]ICFO-Institut de Ciències Fotòniques, The Barcelona Institute of Science and Technology, Av. Carl Friedrich Gauss 3, 08860 Castelldefels (Barcelona), Spain. [4]Instituto de Nanociencia y Materiales de Aragon (INMA), CSIC-Universidad de Zaragoza, 50009 Zaragoza, Spain. [5]Departamento de Fisica de la Materia Condensada, Universidad de Zaragoza, Zaragoza 50009, Spain. [6]Donostia International Physics Center (DIPC) and EHU/UPV, 20018 Donostia-San Sebastián, Spain. [7]IKERBASQUE, Basque Foundation for Science, 48009 Bilbao, Spain. [8]ICREA—Institució Catalana de Recerca i Estudis Avançats, Barcelona 08010, Spain. [9]CIC nanoGUNE BRTA and EHU/UPV, 20018 Donostia-San Sebastián, Spain. ✉e-mail: r.hillenbrand@nanogune.eu

resonators[17-19], however, the rather long infrared wavelengths in dielectric materials prevent the development of deep nanoscale resonator and waveguide structures for the utmost concentration of infrared fields.

Phonon polaritons in polar materials, particularly in two-dimensional (2D) materials[20-24], offer promising opportunities for SEIRA spectroscopy, owing to their long lifetimes paired with ultra-small wavelengths (i.e., confinement). Specifically, hyperbolic phonon polaritons (HPhPs) in hexagonal boron nitride (h-BN) have already proven their utility for SEIRA spectroscopy of nanometer-thin molecular layers[25] and phonon-enhanced mid-IR gas sensing[26]. The high field confinement and lifetimes of HPhPs allow for even achieving vibrational strong coupling of organic molecules with localized[27] and propagating modes[28], as demonstrated with h-BN nanoresonators and unstructured h-BN flakes, respectively. Beyond h-BN, other polar materials such as $MoO_3$[22] and $V_2O_5$[23] exhibit long-lived HPhPs, promising to expand the material basis and spectral ranges of phononic SEIRA spectroscopy in the future.

Phononic SEIRA spectroscopy is based on far-field extinction measurements (Fig. 1a). It thus requires the conversion of HPhPs into photons, which is an inefficient process due to the short HPhP

wavelengths compared to that of photons. Here, we circumvent this problem by introducing on-chip detection of phononic SEIRA (Fig. 1b). Previously, on-chip detection of plasmonic SEIRA has been demonstrated exploiting plasmon polaritons in a hybrid graphene-metamaterial detector array covering a large area of $600 \times 600\ \mu m^2$ [29]. In contrast, our phononic SEIRA implementation not only utilizes HPhPs instead of plasmon polaritons but also employs a single detector as small as $7 \times 4\ \mu m^2$, establishing an efficient and compact SEIRA platform. In our specific demonstration, we deposited organic molecules on a h-BN/graphene/h-BN heterostructure on top of a metal split-gate creating a p–n junction (split-gate detector)[30-32]. The metal split gate concentrates the incident light and launches HPhPs in the heterostructure. These effects together provide strong mid-IR field enhancement at the position of the molecules and the graphene p–n junction (Fig. 1c), enhancing both the molecular vibrational absorption and the photocurrent created at the p–n junction[31-33]. We observe dips in PC spectra at the molecular vibrational frequencies, which, furthermore, are much stronger than the ones observed in a standard far-field transmission experiment, in good agreement with numerical simulations. The sensitivity enhancement increases for thinner molecular layers, owing to the strong confinement of the

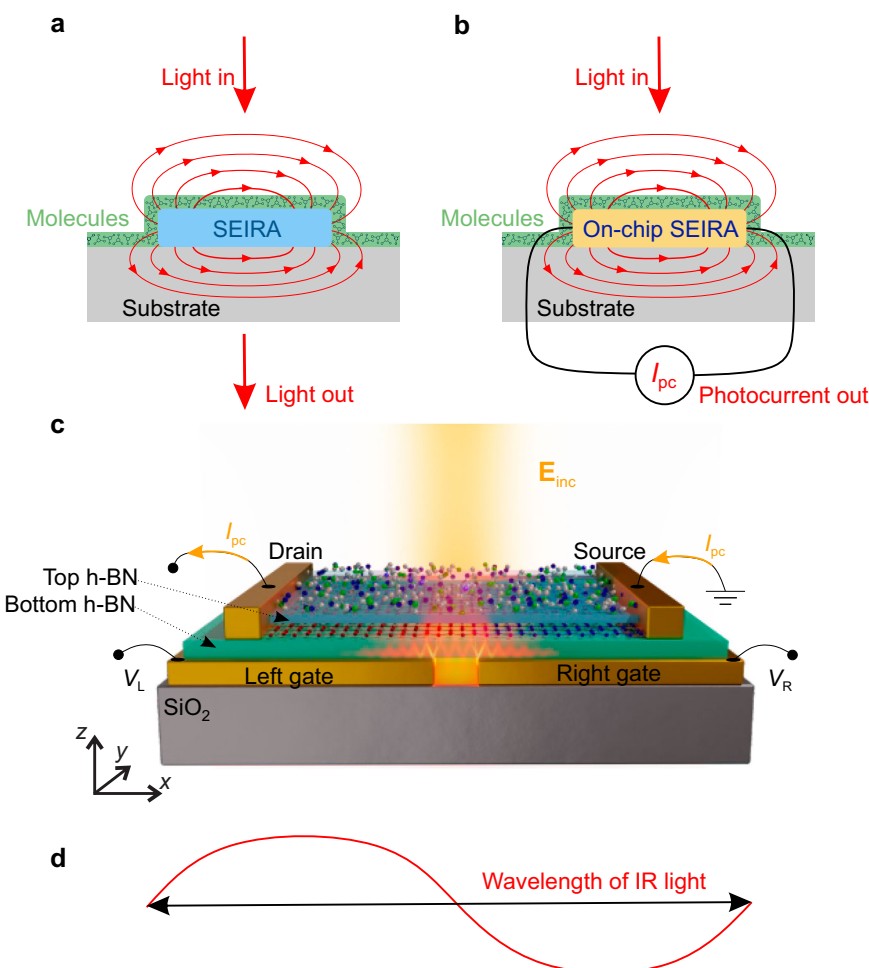

**Fig. 1 | On-chip phononic SEIRA detection of molecular vibrations. a** Far-field SEIRA spectroscopy. Local field enhancement is illustrated by red lines. **b** Sketch of an on-chip SEIRA spectroscopy experiment, where a detector is placed in the near field of a SEIRA structure. **c** Illustration of a graphene split-gate detector with intrinsic mid-IR field enhancement, covered by a thin layer of CBP molecules. The detector consists of a heterostructure comprising an exfoliated monolayer of graphene encapsulated between two thin h-BN layers of 5 and 3 nm thickness. It is placed on top of a metal split gate used for creating a p–n junction in graphene via electric gating. For measuring the photocurrent induced in the p–n junction under illumination, the graphene is contacted by metal electrodes (source and drain). The detector surface is covered by a thin organic layer exhibiting mid-IR molecular vibrations. The size of the detector and thus the maximum size of the probed area of molecules is about $7 \times 4\ \mu m^2 < \lambda_{IR}^2$ (Supplementary Fig. 1), where $\lambda_{IR}$ is about 6.5–7 μm. **d** Red line illustrates an infrared wave with wavelength (indicated by the black double arrow), $\lambda_{IR}$, on the scale of the detector device.

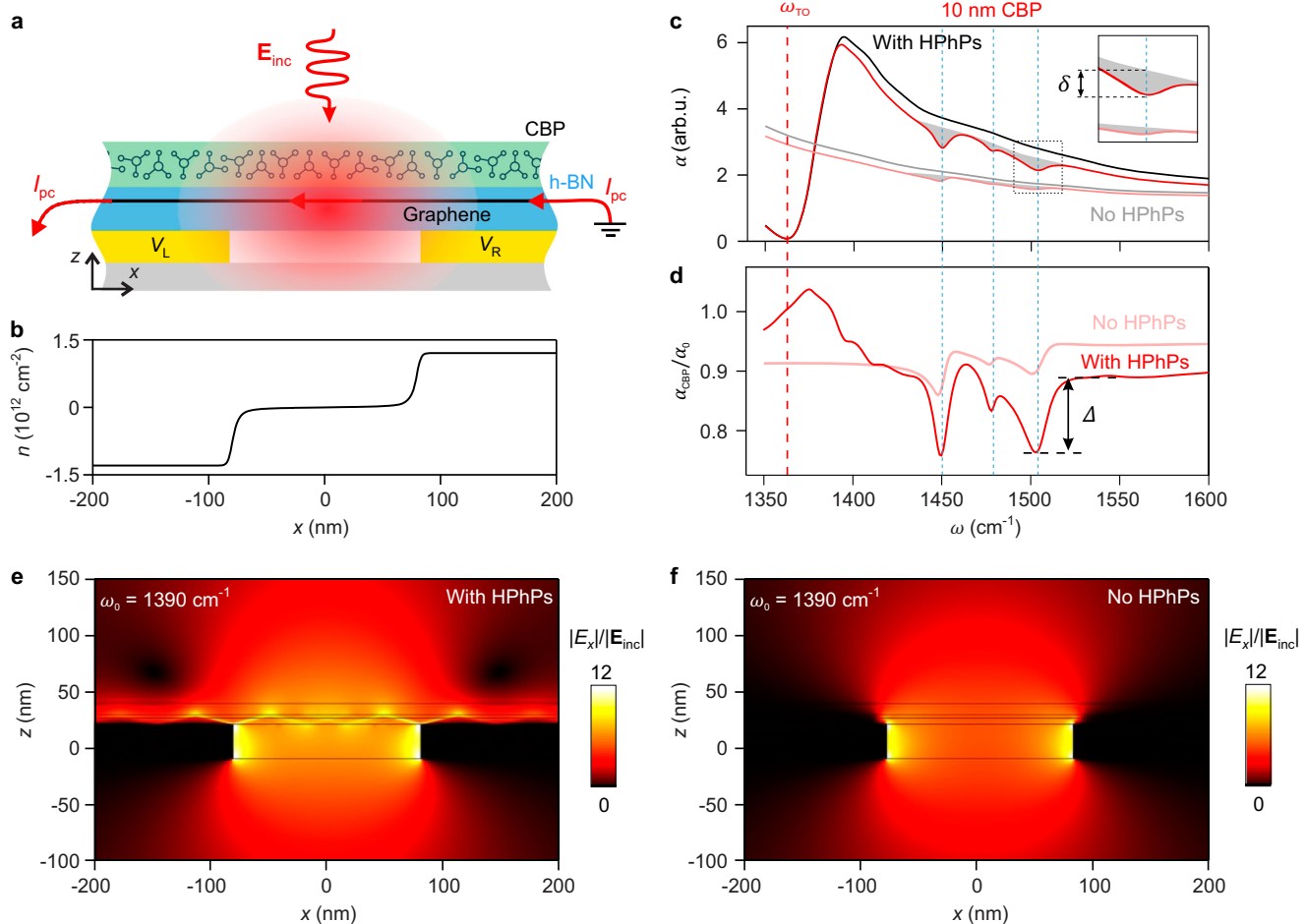

**Fig. 2 | Influence of HPhPs on the sensitivity of our on-chip spectroscopy concept. a** Schematic side view of the gap area of the molecule-covered graphene split-gate detector. **b** Simulated carrier concentration in graphene as a function of lateral position $x$ along the source–drain electrode direction. **c** Simulated mid-IR absorption in graphene (integrated over the whole detector area), which in good approximation is proportional to the photocurrent. The black and red curves show the absorption in graphene without, $\alpha_0$, and with 10 nm thick CBP molecules, $\alpha_{CBP}$, on top of the split-gate detector, respectively. The gray and light red curves show the absorption in graphene in the absence of HPhPs without, $\alpha_0$, and with 10 nm-thick CBP molecules, $\alpha_{CBP}$, on top of the detector, respectively. The vertical dashed red line marks the frequency of the in-plane TO phonon of h-BN. The gray areas indicate the decrease in the absorption due to the molecular-vibrational resonances. The inset shows the zoom-in with indicated depth of the molecular-vibrational resonance dip, $\delta$. **d** The red and light red curves show the normalized absorption in graphene, $\alpha_{CBP}/\alpha_0$, calculated using the data from the panel **b** when there are HPhPs and no HPhPs in the h-BN, respectively. Black arrow indicates the molecular-vibrational contrast, $\Delta$. **c**, **d** Three vertical dashed blue lines indicate the frequencies of the molecular vibrational resonances. **e**, **f** Simulated horizontal $x$-component of the electrical field enhancement, $|E_x|/|\mathbf{E}_{inc}|$, in the gap region of the detector, which is covered by 10 nm-thick molecular layer, in the presence and absence of HPhPs, respectively. Illumination is at $\omega_0 = 1390\ \mathrm{cm}^{-1}$. Source data are provided as a Source Data file.

polariton-enhanced near fields inside and at the surface of the h-BN layers. Our simulations further predict that deep subwavelength-scale areas of thin molecular layers can be detected with enhanced sensitivity as compared to diffraction-limited far-field mid-IR spectroscopy, thus underpinning the potential of our on-chip phononic SEIRA concept to become a compact chemical sensor.

## Results
### Concept and theoretical analysis
Figure 2a illustrates the graphene split-gate detector and its application for on-chip phononic SEIRA detection of molecular vibrations. The detector itself consists of a heterostructure comprising a mono-layer of exfoliated graphene that is encapsulated between two h-BN layers and placed on top of two metal gates separated by a narrow gap of about 160 nm (see fabrication details in the "Methods" section and Supplementary Note 1). By applying voltages of opposite signs to the gates (0.3 V to the left gate and −0.28 V to the right gate), a p–n junction is created in graphene in a small region above the gap. Fig. 2b illustrates the p–n junction by showing a simulated carrier

concentration profile across the gap (Supplementary Note 5.1). When the detector is illuminated by mid-IR light, the graphene p–n junction is heated up, and via the photo-thermoelectric effect, a photovoltage $V_{PC}$ is generated between the source and drain contacts[31–33] (see the electrical and optical detector characterization in Supplementary Notes 2, 4). Importantly, the metal gates concentrate the incident light into the gap area through the lightning-rod effect. The enhanced fields at the gate edges launch HPhPs (manifested by bright rays in the simulated field distribution shown in Fig. 2e) in the h-BN/graphene/h-BN heterostructure, which exist between the transverse and longitudinal optical phonon frequencies of h-BN, $\omega_{TO} = 1363\ \mathrm{cm}^{-1}$ and $\omega_{LO} = 1617\ \mathrm{cm}^{-1}$, where the in-plane permittivity of h-BN is negative[20,21]. Both of these effects strongly increase the mid-IR field enhancement in the graphene p–n junction and, thus, the absorption in graphene in a small region above the gap area, subsequently enhancing the photo-voltage $V_{PC}$ between the source and drain contacts[31–33]. In Fig. 2c, we illustrate the spectral response of the detector with a simulated spectrum of the absorption in graphene, $\alpha_0$, (black curve), which in good approximation describes the photovoltage[31,32] (see the

simulation details in the "Methods" section and Supplementary Note 5). We clearly observe an absorption maximum slightly above the $\omega_{TO}$ (indicated by the vertical dashed red line), which can be attributed to localized resonant HPhP mode in the gap between the gates (further discussion see below and in Supplementary Note 6).

To demonstrate that the graphene split-gate detector can be used for on-chip detection of molecular vibrations, we simulated the absorption spectrum of graphene, $\alpha_{CBP}(\omega)$, when a 10-nm-thick layer of 4,4′-bis($N$-carbazolyl)-1,1′-biphenyl (CBP) molecules are placed on top of the detector (as illustrated in Fig. 2a). Fig. 2c and d show $\alpha_{CBP}$ and the normalized spectrum, $\alpha_{CBP}/\alpha_0$, (red curves), respectively, where $\alpha_0$ is the reference (i.e., background) spectrum of the bare detector. This normalization procedure is analog to the one used in standard spectroscopy experiments to eliminate the spectral characteristic of the light source, spectrometer and detector. We note that the small dips in $\alpha_{CBP}/\alpha_0$ around 1400 cm$^{-1}$ in the spectra when HPhPs are present in the h-BN[27,34] are attributed to the higher-order slab modes in the h-BN[27,34]. These modes exhibit distinct redshifts due to the presence of the thin molecular layer on top of the detector thus leading to small variations in the normalized spectra. Importantly, in both spectra, we can clearly observe dips at the molecular vibrational frequencies (marked by three vertical dashed blue lines).

To elucidate the impact of HPhPs on the sensitivity of our on-chip molecular-vibrational spectroscopy concept, we calculated the absorption spectra of graphene in the absence of HPhPs (gray and light red curves in Fig. 2c, d; see the "Methods" section). In this case, the bright rays in the field distribution vanish (Fig. 2e). In the non-normalized spectra (represented by the gray and light red curves in Fig. 2c), we observe a relatively flat spectral profile and a generally reduced photocurrent. Particularly, we see that the maximum slightly above $\omega_{TO}$ (indicated by the vertical dashed red line) is not present. Further, the depth of the molecular-vibrational resonance dips, $\delta$, is reduced (light red curve in Fig. 2c). We conclude that HPhPs do not only enhance the absolute photocurrent but also the photocurrent signatures associated with the molecular-vibrational resonances of the molecular layer on top of the detector (see the additional simulation in Suppl. Note 6). Interestingly, we also observe that the depth of the molecular-vibrational resonance dip in the normalized spectrum (i.e., the molecular-vibrational contrast), $\Delta$, is enhanced by the HPhPs (Fig. 2d). Note that, generally, the enhancement of a molecular-vibrational contrast cannot be explained by merely an increased detector sensitivity, but by an enhanced light-molecule interaction mediated by a strong optical field enhancement at surfaces or nanostructures in proximity to the molecules. We thus explain the enhancement of $\Delta$ in our specific on-chip spectroscopy concept by the interaction of the molecules on top of the detector with the strong near fields of the h-BN HPhPs.

### Experimental demonstration

For an experimental demonstration of on-chip phononic SEIRA detection of molecular vibrations, we implemented the graphene split-gate detector (bare and covered with CBP molecules) into a custom-made Fourier transform (FT) spectrometer, where it replaced the standard IR detector (see the "Methods" section and Supplementary Note 3). As a light source, we used a mid-IR continuum laser covering the spectral range from 1200 to 1800 cm$^{-1}$ with an average power of about 0.6 mW. We applied a voltage of $V_L = 0.15$ V to the left gate and a voltage of $V_R = -0.4$ V to the right gate of the graphene split-gate detector, where the maximum spectrally integrated photocurrent was obtained (Supplementary Note 4). The frequency-resolved photocurrent, $I_{PC}(\omega)$, was obtained by standard FT spectrometer operation and interferogram processing (see the "Methods" section). We first measured the PC spectrum of the bare detector, $I_{PC}^0$, which serves as a reference (i.e. background) spectrum to obtain the normalized PC spectra of the molecule-covered detector, $I_{PC}^{CBP}/I_{PC}^0$, where the spectral response characteristics of the laser, spectrometer and bare detector are eliminated. After the reference measurement, we evaporated (see the "Methods" section) a 10-nm-thick layer of CBP molecules directly on top of the detector (as shown in the sketch in Fig. 3a) and measured the PC spectrum, $I_{PC}^{CBP}$. Next, we repeatedly evaporated CBP molecules on top of the detector and measured the PC spectrum, until a CBP layer thickness of 100 nm was reached. The red, blue, and green curves in Fig. 3b show the normalized PC spectra, $I_{PC}^{CBP}/I_{PC}^0$, for the 10, 40, and 100 nm-thick CBP layers, respectively (the complete set of spectra is shown in Supplementary Fig. 13). In all spectra, we observe a pronounced peak around 1370 cm$^{-1}$ (indicated by the gray circle), followed by a photocurrent that tends to increase with increasing frequency. Further, we observe three dips in the PC spectra (marked by vertical dashed blue lines), whose depths increase with the thickness of the CBP layer. Most importantly, the spectral dip positions correspond to the frequencies of molecular vibrations of CBP, which can be recognized as peaks in the imaginary part of the dielectric function of CBP[25] (Fig. 3d), demonstrating that molecular vibrational fingerprints of nanometer-thin molecular layers of lateral sizes around $\lambda_{IR}$ can be detected. The experimental spectra are qualitatively well reproduced by calculated normalized absorption spectra, $\alpha_{CBP}/\alpha_0$ (Fig. 3c), apart from a slightly rising background observed only experimentally. This discrepancy may stem from assuming a 2D geometry for the detector in the simulations, that is, the detector is considered infinite in the $y$-direction.

In contrast to resonant SEIRA spectra, where typically Fano-type line shapes are observed[3], the spectral dips occur at the molecular vibrational resonances (marked by vertical dashed lines) and in the simulations exhibit nearly symmetric line shapes. We attribute this observation to the simultaneous coupling of the molecular vibrations to a variety of resonant, non-resonant modes and dark polariton modes (the latter not observed in far-field SEIRA spectroscopy), yielding in average dip-like line shapes. Further systematic studies are certainly needed for in-depth clarification of this interesting phenomenon, which, however, goes beyond the scope of this work.

We also note that in both experiment and calculation we observe a peak around 1370 cm$^{-1}$ and a baseline, which are not related to the molecular vibrational resonances. They are caused by the slight frequency shift of the absorption spectrum of the molecule-covered detector compared to that of the bare detector, which can be seen by comparing the red and black curves in Fig. 2c. This shift can be explained by the background permittivity of the CBP molecules, $\varepsilon_\infty = 2.8$ (see Supplementary Note 7 for additional simulations), acting as a dielectric load on the detector. The detector, thus, can also be applied for pure refractive-index sensing.

### Comparison with far-field infrared transmission spectroscopy

For evaluating the spectroscopy benefit of placing the molecular layer directly on top of the graphene split-gate detector, we compare the PC spectra of Fig. 3b with standard far-field FTIR transmission spectra of equally thick CBP layers (Fig. 3f). To that end, we deposited CBP molecules onto a CaF$_2$ substrate and placed them in a distance of more than 10 cm in front of the bare graphene split-gate detector (illustrated in Fig. 3e). Note that for this experiment we used the same graphene split-gate detector used for recording the spectra of Fig. 3b but before any molecule layer was deposited onto the detector itself (i.e. before recording the spectra of Fig. 3b). The normalized transmission spectra $T_{CBP}/T_0$ (where $T_0$ is the reference spectrum obtained from transmission through the bare CaF$_2$ substrate) show the typical absorption dips at the molecular vibrational frequencies (marked by vertical dashed lines), in excellent agreement with calculated normalized transmission spectra (Fig. 3g) (see the "Methods" section). However, the dip depths in the transmission spectra in Fig. 3f, g are much less pronounced compared to Fig. 3b, c, particularly for thin layers. Remarkably, for the 10 nm thin CBP layer, we observe the molecular

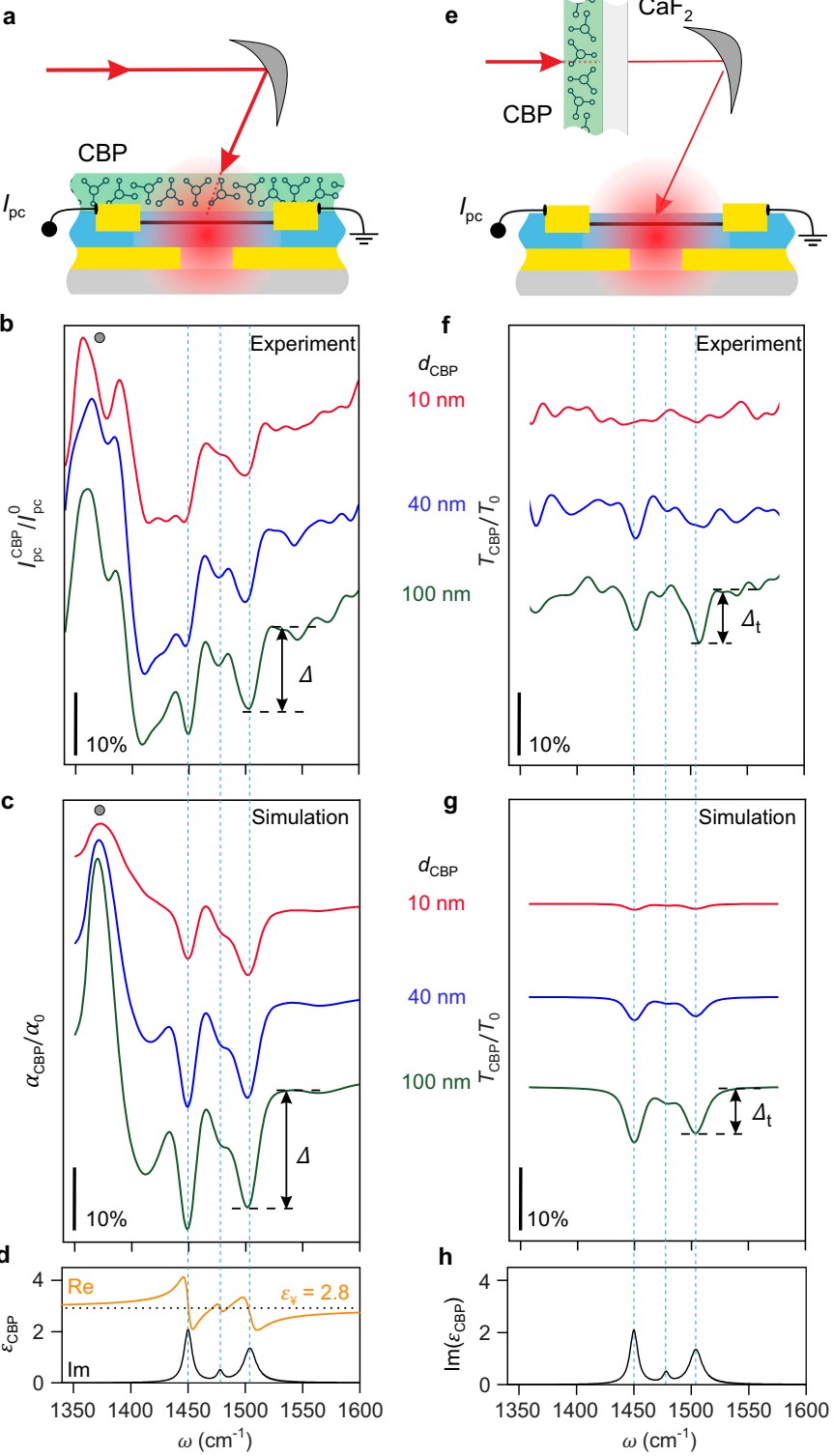

**Fig. 3 | Polariton-enhanced near-field photocurrent spectroscopy vs. far-field transmission FTIR spectroscopy. a** Schematic side view of the molecule-covered graphene split-gate detector. **b** Experimental PC spectra for differently thick CBP layers deposited on top of the detector. All spectra are normalized to the PC spectrum of the bare detector. **c** Simulated normalized absorption in graphene for different thicknesses of CBP layers placed directly on top of the detector. All spectra are normalized to the absorption in graphene without molecules on top of the detector. **d** Real and imaginary parts of the dielectric function of CBP (orange and black curves, respectively). The three vertical blue dashed lines indicate the resonance frequencies of the molecular vibrations of CBP. **e** Schematic side view of the bare graphene split-gate detector illuminated through a molecule-covered CaF$_2$ substrate. **f** Experimental transmission spectra of the bare detector when differently thick CBP layers on a CaF$_2$ substrate are placed in front of the detector at a centimeter distance. All spectra are normalized to the transmission spectrum obtained with a bare CaF$_2$ substrate. **g** Simulated normalized transmission spectra through differently thick CBP layers on a CaF$_2$ substrate. **h** Same as panel **d** but without the real part of the dielectric function. **c, g** All simulated spectra were convoluted (see the "Methods" section) such that the spectral resolution corresponds to that of the experimental spectra, that is, 6.5 cm$^{-1}$ (original simulated spectra shown in Supplementary Note 8). Arrows and symbols $\Delta$ and $\Delta_t$ illustrate how the molecular-vibrational contrasts shown in Fig. 4 were measured. Source data are provided as a Source Data file.

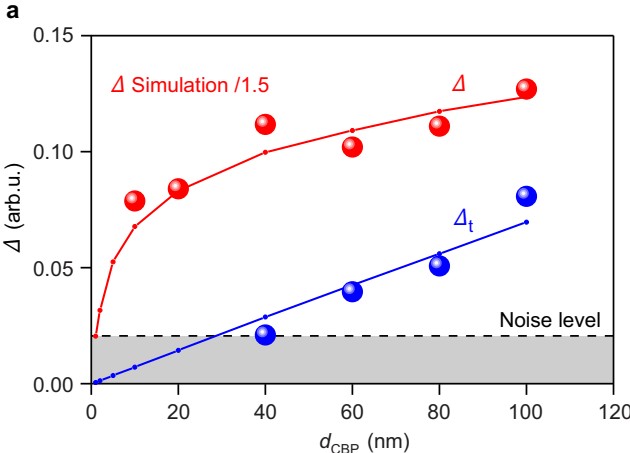

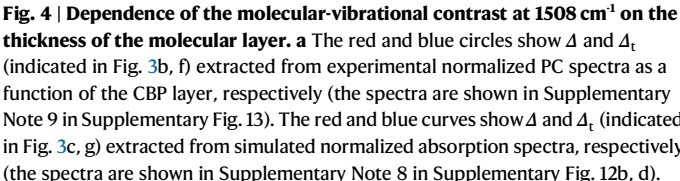

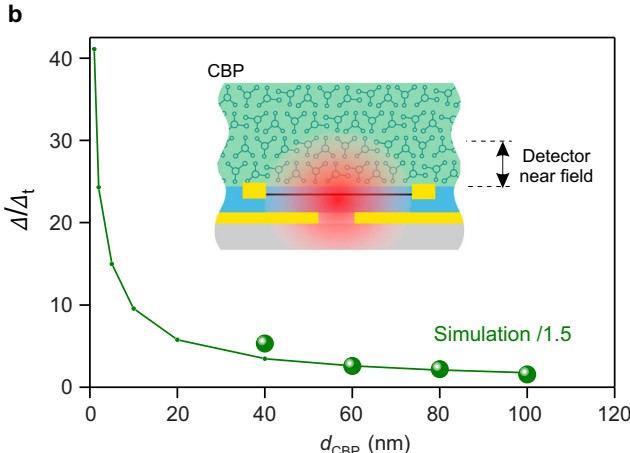

**Fig. 4 | Dependence of the molecular-vibrational contrast at 1508 cm⁻¹ on the thickness of the molecular layer. a** The red and blue circles show $\Delta$ and $\Delta_t$ (indicated in Fig. 3b, f) extracted from experimental normalized PC spectra as a function of the CBP layer, respectively (the spectra are shown in Supplementary Note 9 in Supplementary Fig. 13). The red and blue curves show $\Delta$ and $\Delta_t$ (indicated in Fig. 3c, g) extracted from simulated normalized absorption spectra, respectively (the spectra are shown in Supplementary Note 8 in Supplementary Fig. 12b, d).

**b**, The green circles and curve show the enhancement of molecular-vibrational contrast, $\Delta/\Delta_t$, calculated from the experimental and simulated values from the panel a, respectively. The inset shows that when a thick molecular layer is placed on top of the detector, only a portion of the molecules interact with the near fields of the detector. **a,b**, The values of the simulated red and green curves were divided by 1.5 for visual comparison with the experimental data. Source data are provided as a Source Data file.

vibrational fingerprint only when the layer is deposited directly onto the detector, clearly demonstrating the superior sensitivity due to the polariton-enhanced near fields in the vicinity of the split gate detector. It is important to note that the same detector was used for both experiments, ensuring that the key detector characteristics, such as sensitivity, detectivity, and signal-to-noise ratio, remained constant. As a result, it allows for a reliable comparison and verification of the superior performance of the on-chip phononic SEIRA detection of molecular vibrations compared to a standard far-field approach employing the same detector.

To quantify the enhancement of the molecular-vibrational contrast when the molecules are placed directly on top of the detector, we measured the depth of the molecular-vibrational contrast at 1508 cm⁻¹ in the on-chip phononic SEIRA and far-field FTIR transmission spectra (denoted $\Delta$ and $\Delta_t$, respectively, and illustrated in Fig. 3b, f), and plotted them in Fig. 4a as a function of the layer thickness $d_{CBP}$ (the full data set of spectra are shown in Supplementary Note 9 in Supplementary Fig. 13). In Fig. 4b we show the ratio $\Delta/\Delta_t$ (i.e. the enhancement of molecular-vibrational contrast). We find that $\Delta$ (red symbols in Fig. 4a, on-chip phononic SEIRA experiment) is enhanced by a factor of two compared to $\Delta_t$ (blue symbols in Fig. 4a, far-field experiment) at $d_{CBP} = 100$ nm, increasing to a factor of about 5 at $d_{CBP} = 40$ nm (Fig. 4b). Most important, for the 10 and 20 nm-thick molecular layers we can obtain measurable values only for $\Delta$ but not for $\Delta_t$. The enhancement of $\Delta$ relative to $\Delta_t$ is qualitatively confirmed by numerical simulations (solid lines in Fig. 4a and b; for details, see the "Methods" section). For quantitative agreement, we have to divide the simulated $\Delta$-values by a factor of 1.5, which we attribute to the approximations made in the simulations, such as assuming a plane wave illumination and a 2D detector geometry (i.e., the detector is infinite in the y-direction, see Supplementary Note 5). For molecular layers with $d_{CBP} < 40$ nm, the simulations predict large enhancements of molecular-vibrational contrast, $\Delta/\Delta_t$, reaching well above one order of magnitude for $d_{CBP} < 10$ nm (Fig. 4b), which we explain by the strong field concentration in the vicinity of the detector surface (illustrated in the inset of Fig. 4b).

**Sensitivity to molecular layers with nanoscale lateral size**
Finally, we predict on-chip phononic SEIRA detection of deep subwavelength-scale wide molecular layers (i.e., molecular

stripes). To that end, we calculated on-chip phononic SEIRA and far-field transmission spectra, $\alpha_{CBP}/\alpha_0$ (Fig. 5a) and $T_{CBP}/T_0$ (Fig. 5b), respectively, for 300 nm-wide (about twice the gap width) molecular stripes of different thicknesses placed either directly above the detector´s p–n junction (illustrated by the inset of Fig. 5a) or on a CaF₂ substrate (illustrated by the inset of Fig. 5b). For a conservative estimation of the far-field transmission spectra, we assumed a plane wave illumination on a width of 3 μm (corresponding to a diffraction-limited focus diameter). For all thicknesses, we observe a larger enhancement of the molecular-vibrational contrast, $\Delta/\Delta_t$, compared to Fig. 3. For example, we find $\Delta/\Delta_t = 80$ for the 10 nm-thick stripe (compare red spectra in Fig. 5a, b), which is 5 times larger than for the 10 nm-thick layer (compare red spectra in Fig. 3c, g). We attribute this additional contrast enhancement to the strong localization of the polaritonic field enhancement near the p–n junction. As a consequence, when the width of the molecular layer decreases, the molecular vibrational absorption contrast, $\Delta$, is less reduced compared to that observed in far-field transmission spectroscopy, $\Delta_t$, where the IR field distribution is considered to be homogeneous. These simulations demonstrate the extraordinary sensitivity of on-chip phononic SEIRA detection to molecular stripe, surpassing the limitations of conventional diffraction-limited far-field transmission spectroscopy.

## Discussion
We note that the sensitivity of on-chip SEIRA detection has not yet reached the state-of-the-art far-field resonant SEIRA experiments utilizing nitrogen-cooled mercury cadmium telluride (MCT) detector, where a sensitivity of about 500 molecules has been demonstrated[35]. This can be attributed to the use of a non-resonant split-gate design, the mismatch between the HPhP resonance localized in the gap of split-gate and the molecular vibrational resonances, and the lower sensitivity of the room temperature graphene split-gate detector compared to cooled MCT detectors. Future optimization of the device design for resonant detection could significantly boost sensitivity. This optimization includes adjusting the h-BN layer thicknesses, modifying bottom gate lengths, and fine-tuning the split-gate gap size. Importantly, the room-temperature operation and the complementary

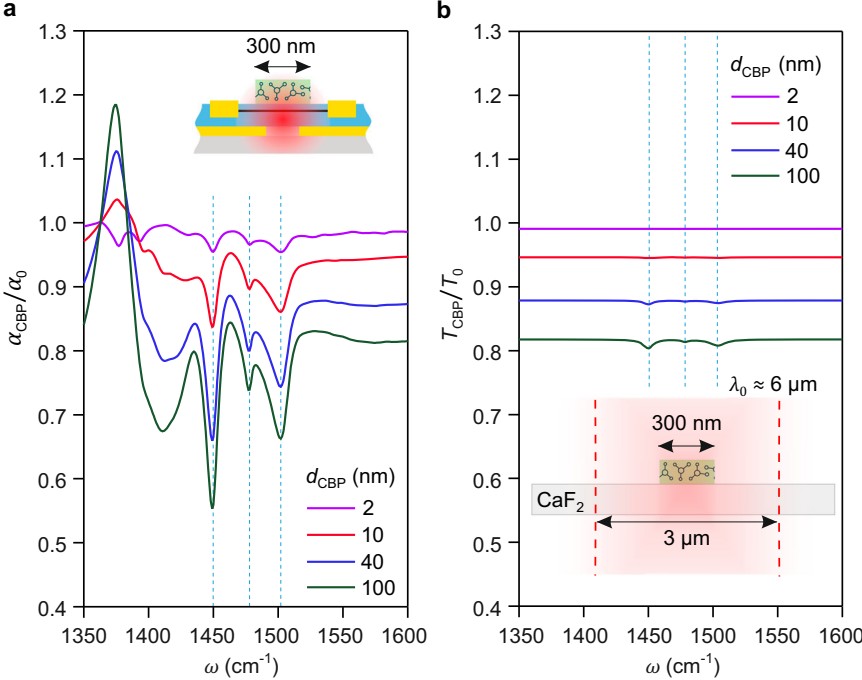

**Fig. 5 | Theoretical sensitivity to the subwavelength-scale area of the thin layers of molecules. a** Simulated normalized graphene absorption spectra for a 300 nm-wide stripe and different thicknesses of CBP molecules placed on top of the p–n junction of the detector. Inset shows the schematic side view of the graphene split-gate detector partially covered by the molecular stripe. **b** Simulated normalized transmission spectra through the 300 nm-wide stripe and different thicknesses of CBP molecules placed on top of the CaF$_2$ substrate, assuming the 3 μm-wide light beam. Curves are shifted vertically for clarity. Inset shows the schematic side view of the molecular stripe in the mid-IR focus with a waist of 3 μm. Note that in contrast to Fig. 3, the simulated spectra are not convoluted. Source data are provided as a Source Data file.

metal-oxide-semiconductor (CMOS) compatibility of graphene detectors[36] represent progress toward compact on-chip sensors functioning under ambient conditions.

Our work demonstrates that the near fields above the p–n junction of a graphene split-gate detector—which is enhanced by h-BN HPhPs—can be used for the mid-IR on-chip phononic SEIRA detection of wavelength-size nanometer-thin layers of organic molecules deposited directly on top of the detector. We observe dips in the PC spectra at the molecular vibrational frequencies, which are much stronger than those measured in standard far-field FTIR transmission experiments, confirming the enhanced sensitivity of on-chip phononic SEIRA detection to nanometer-thin molecular layers. Numerical simulations predicted a significant further sensitivity enhancement compared to standard far-field transmission spectroscopy when the lateral size of the molecular layer is reduced to a deep sub-wavelength scale.

In the future, the applicability of on-chip phononic SEIRA detection can be expanded to a wider frequency range and different analytes by employing several approaches. One approach could be the combination of different phononic van der Waals (vdW) materials in heterostructures for multispectral applications[37], each material supporting phonon polaritons in its own Reststrahlen bands (RBs). Corresponding materials include MoO$_3$ (three RBs between 600 and 1000 cm$^{-1}$)[22,38], V$_2$O$_5$ (three RBs between 500 and 1000 cm$^{-1}$)[23], and LiV$_2$O$_5$ (three RBs between 950 to 1025 cm$^{-1}$)[39]. Further, thin layers of conventional 3D materials such as SiC[40] could be incorporated. We also envision exploiting plasmon polaritons in the detector´s active graphene layer or by incorporating additional graphene layers[41]. The molecule-covered detector may also be used for the PC near-field spectroscopy[42]. Furthermore, the detector can be potentially combined with a transparent microfluidic system for biosensing in aqueous solutions[43]. Moreover, nano- and Å-scale channels fabricated with vdW materials[44,45] could also be integrated above the graphene p–n junction

to enable aqueous molecular on-chip sensing. Finally, we envision either integrating tunable quantum cascade lasers on the same chip or enabling the electrical generation of polaritons[46,47] in the various layers comprising the detector, which could lead to the development of compact, fully on-chip phononic SEIRA spectroscopy devices.

## Methods

### Detector fabrication
First, the metallic split-gate on a Si/SiO$_2$ substrate was fabricated via electron beam lithography (EBL). To that end, 300 nm of PMMA resist was spin-coated onto the substrate. After EBL patterning and subsequent development of the resist in a solution of methyl isobutyl ketone (MIBK)/isopropanol (IPA), a 2 nm-thick Ti layer and an 8 nm-thick Au layer were thermally evaporated onto the sample, followed by a lift-off procedure in acetone[48]. The dimensions of the fabricated split-gate were determined from the scanning electron microscopy (SEM) image shown in Suppl. Fig. 1b and summarized in Supplementary Fig. 1a.

Second, graphene was encapsulated between two thin layers of h-BN. To that end, two flakes of h-BN and single-layer graphene were mechanically cleaved and exfoliated onto freshly cleaned Si/SiO$_2$ substrates. Then, a heterostructure stack (h-BN/graphene/h-BN) was fabricated using the vdW assembly technique[48–50] and placed onto the metal split-gate. The thicknesses of the top and bottom h-BN layers were 3 and 4.5 nm, respectively (measured by atomic force microscope (AFM)).

Third, the graphene was structured into a rectangular shape to avoid sample inhomogeneity and reduce the potential leakage between the source–drain electrodes with the local gates. To that end, a 300 nm-thick PMMA layer was spin-coated on top of the h-BN/graphene/h-BN heterostructure on top of the split gate. Then, EBL was used to pattern an etching mask with a subsequent development step of the PMMA in the MIBK/IPA solution. The top h-BN was selectively

etched using $SF_6$ gas for 1 min at a pressure of ≈40 cm³ STP min⁻¹ [32]. The graphene was etched using $O_2$ gas at a pressure of ≈30 cm³ STP min⁻¹ for ≈1 min.

Finally, metal contacts (for measuring the photocurrent in the graphene layer) were added. To that end, 300 nm of PMMA resist was spin-coated onto the etched sample. Source and drain contacts were patterned using EBL, followed by resist development and etching of h-BN and graphene (same as described above). Then, a 5 nm-thick layer of Cr and a 60 nm-thick layer of Au were evaporated. The process was completed with a lift-off in acetone.

Before using the detector for photocurrent measurements, we performed a mechanical cleaning technique known as "brooming"[51], typically used to remove lithographic residues, such as PMMA, from the surface. Brooming is done by scanning an AFM tip in contact mode across the sample area to be cleaned, here the p–n junction of the detector. Optical and AFM images of the broomed detector are shown in Supplementary Fig. 1c, d, respectively.

The electrical characterization of the fabricated detector is described in Supplementary Note 2.

## Molecule deposition

4,4′-bis(*N*-carbazolyl)-1,1′-biphenyl (CBP) of sublimed quality (99.9%) (Sigma-Aldrich, Saint Louis, MO, USA) was thermally evaporated in a high-vacuum evaporator chamber (base pressure < 10⁻⁹ mbar), at a rate of 0.1 nm s⁻¹ using a Knudsen cell. To prevent the agglomeration of molecules into islands during the initial growth of molecular layers within the first 10 nm, the Si/$SiO_2$ substrate with the graphene split-gate detector and bare $CaF_2$ substrate, respectively, were placed onto the cold finger, which was cooled down to a temperature of -77–80 K by liquid nitrogen.

## Frequency-resolved photocurrent measurement using the Fourier transform spectroscopy principle

A schematic of the spectroscopy setup described in the following is shown in Supplementary Fig. 3b. The graphene split-gate detector was illuminated by a broadband mid-IR laser (supercontinuum laser, Femtofiber pro-IR from Toptica, Gräfelfing, Germany) via a parabolic mirror with a numerical aperture of about 0.3. The laser was tuned to emit in the frequency range 1200–1700 cm⁻¹ with an average power of about 0.6 mW. The PC signal from the graphene split-gate detector was measured with a low-noise current amplifier with a gain factor of 10⁶ (DLPCA-200 from Femto, Germany). The gate voltages were applied using a DAQ card (USB−6218 from National Instruments, USA). For frequency-resolved PC measurements, we employed the FT spectroscopy principle: the illumination of the detector was done via a Michelson interferometer comprising a ZnSe beamsplitter and two planar mirrors. The position of one mirror (scanned mirror) was controlled by a piezo-electric scanner. The PC signal was recorded as a function of the mirror position, yielding interferograms. Each interferogram had a maximum length of 0.8 mm. For the apodization of the interferograms, a Hann window function was applied. After zero-filling (4× padding), the interferograms were Fourier transformed to obtain the PC spectra with a spectral resolution of about 6.5 cm⁻¹.

We first used the setup described above to measure the IR transmission spectra of thin CBP molecule layers on a double-sided polished $CaF_2$ substrate (Fig. 3f and Supplementary Fig. 13b), which can be considered as conventional FTIR transmission spectra. To that end, a 10 nm-thick layer of CBP was deposited onto one-half of the $CaF_2$ substrate (see the "Methods" section) by using a shadow mask to protect the other half from molecular deposition. After depositing the molecules, the $CaF_2$ substrate was placed into the beam path of our spectroscopy setup, positioning it in front of the parabolic mirror (i.e., outside the interferometer). First, we measure the reference transmission spectrum by aligning the substrate such that all laser radiation passes through the bare (uncoated) half of the substrate. Then, we

moved the substrate such that the laser radiation passed through the molecule-covered half of the $CaF_2$ substrate, enabling us to measure the transmission spectrum of a thin layer of CBP molecules. The deposition/spectroscopy cycle was repeated 5 times, ensuring that one half of the substrate remained bare (uncoated), until a cumulative CBP layer thickness of 100 nm was reached. The thickness of the deposited layer in the first cycle was 10 and 20 nm in subsequent cycles. Each spectrum is shown in Fig. 3f and Supplementary Fig. 13b was obtained by FT of an average of 20 interferograms recorded with 1024 pixels (positions of the scanned mirror) and an integration time of 25 ms per pixel.

For the on-chip SEIRA experiments (Fig. 3b and Supplementary Fig. 13a), we first measured the PC spectrum of the bare detector, which served as the reference spectrum for all subsequent measurements. Then, the detector was removed from the setup and placed into the evaporation chamber for the deposition of a 10 nm-thick layer of CBP molecules directly onto the detector surface (see above). After deposition, the detector was placed back into the spectroscopy setup to measure the PC spectrum of the molecule-covered detector. The deposition/spectroscopy cycle was repeated five times until a CBP layer thickness of 100 nm was reached. The thickness of the deposited layer in the first cycle was 10 and 20 nm in subsequent cycles. Each spectrum shown in Fig. 3b and Supplementary Fig. 12a was obtained by FT of an average of 20 interferograms recorded with 1024 pixels (positions of the scanned mirror) and an integration time of 20 ms per pixel.

For the measurements shown in Supplementary Note 4 (Supplementary Fig. 4a, b), the position of the scanning mirror of the interferometer was fixed, yielding a spectrally integrated photocurrent. The broadband laser was chopped, and the output of the current amplifier was connected to a lock-in amplifier (7280 Perkin Elmer, USA). The photocurrent $I_{PC}$ was derived from the output signal of the lock-in amplifier, $V_{LIA}$, using the equation $I_{PC} = \frac{2\pi\sqrt{2}}{4\xi} V_{LIA}$ (see refs. 32,52), where $\xi = 10^6$ is the gain factor of the current amplifier in AV⁻¹.

We note that the setup was implemented with optical, electronic and hardware components of a commercial scattering-type scanning near-field optical microscope (s-SNOM) comprising a nano-FTIR module (Neaspec/Attocube, Germany).

The commercial nano-FTIR module consists of a ZnSe beamsplitter and one planar mirror (scanned mirror), whose position is controlled by a piezo-electric scanner. In the standard s-SNOM setup, as shown in Supplementary Fig. 3a, the light from the laser is split into two beams (reference and illumination beams) by a ZnSe beamsplitter. The reference beam is reflected in a scanned mirror. The illumination beam is focused on an AFM tip by a parabolic mirror, where the illumination beam is then scattered. At an HgCdTe (MCT) detector, the light backscattered from the AFM tip interferes with the back-reflected reference beam. The MCT detector signal is recorded as a function of the position of the scanned mirror that is linearly translated.

To perform our frequency-resolved photocurrent measurement, we modified the setup by removing the MCT detector and the AFM tip, rotating the nano-FTIR module, and adding a second planar mirror to build the Michelson interferometer, as shown in Supplementary Fig. 3b.

## Numerical simulations

Full-wave numerical simulations using the finite-element method (COMSOL) were performed to simulate the electrical field enhancement in the gap region of the detector and to calculate the absorption in graphene (see Supplementary Note 5). The dielectric permittivities of h-BN, Au, CBP, $SiO_2$, and Si are provided in Supplementary Note 5.2.

The simulations shown in Fig. 2, which were performed in the absence of HPhPs, employed a frequency-independent permittivity tensor for h-BN, where $\varepsilon_{h-BN}^{\perp}(\omega) = \varepsilon_{\infty}^{\perp} = 4.98$ and $\varepsilon_{h-BN}^{\parallel}(\omega) = \varepsilon_{\infty}^{\parallel} = 2.95$.

## Data availability

The data that support the findings of this study are available from the corresponding author upon request. The raw interferograms generated in this study are available from Zenodo[53]. Source data are provided with this paper.

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

## Acknowledgements

The work was financially supported by the European Union's Horizon 2020 research and innovation program under grant agreement Nos. 785219 and 881603 (GrapheneCore2 and GrapheneCore3 of the Graphene Flagship). R.H., F. Casanova, and L.E.H. acknowledge funding by the Spanish MICIU/AEI/10.13039/501100011033 (grant CEX2020-001038-M). R.H. acknowledges funding by the Spanish MICIU/AEI/10.13039/501100011033 and ERDF/EU (grants RTI2018-094830-B-I00 and PID2021-123949OB-I00). F. Casanova and L.E.H. acknowledge funding by the Spanish MICIU/AEI/10.13039/501100011033 and ERDF/EU (grant PID2021-122511OB-I00). F.H.L.K. acknowledges funding by the Spanish MICIU/AEI/10.13039/501100011033 (grants FIS2016-81044, PID2019-106875GB-I00, CEX2019-000910-S, PCI2021-122020-2A and PDC2022-133844-100). A.N. acknowledges funding by the Spanish MICIU/AEI/10.13039/501100011033 and ERDF/EU (grants PID2020-115221GB-C42 and PID2023-147676NB-I00); by the Department of Education of the Basque Government (grant PIBA-2023-1-0007). L.M.-M and T.S. acknowledge funding by the Spanish MICIU/AEI/10.13039/501100011033 (grants PID2020-115221GB-C41 and CEX2023-001286-S); by the Government of Aragon through Project Q-MAD. F.H.L.K, S.C., and V.P. acknowledge funding by the European Union (ERC, POLARSENSE, 101123421). Views and opinions expressed are, however, those of the author(s) only and do not necessarily reflect those of the European Union or the European Research Council Executive Agency. Neither the European Union nor the granting authority can be held responsible for them.

## Author contributions

R.H. conceived the initial study and supervised the work. S.C. fabricated the detector with the help of V.-V.P. supervised by F.H.L.K. A.B. performed the experiments and data analysis. S.C. participated in the experiments and data analysis. M.A. contributed to the initial experiments. F. Calavalle performed molecular deposition supervised by F. Casanova and L.E.H. K.D., T.M.S., and A.B. performed numerical simulations supervised by L.M.-M and A.Y.N. A.B. and R.H. wrote the manuscript with input from S.C., T.M.S., L.M.-M., A.Y.N., and F.K. All authors contributed to the scientific discussion and paper revisions.

## Competing interests

R.H. is a co-founder of Neaspec GmbH, now part of Attocube AG, a company producing scattering-type scanning near-field optical microscope systems, such as the one described in this work. Apart from this, the authors declare no competing interests.
