## [Transparent Peer Review file · Nature Communications]

On-chip phonon-enhanced IR near-field detection of molecular vibrations

Corresponding Author: Professor Rainer Hillenbrand

This manuscript has been previously reviewed at another journal. This document only contains reviewer comments, rebuttal and decision letters for versions considered at Nature Communications.

Version 0:

Reviewer comments:

Reviewer #1

(Remarks to the Author)

In the revised manuscript, the authors have included additional discussion on the underlying mechanisms, comparisons with other MIR detectors, particularly graphene-based detectors, and detailed discussions on how to extend the response bandwidth in future iterations. These additions significantly enhance the clarity and help readers evaluate the real performance and potential of the method. While some technical challenges remain unresolved, as a demonstration of the first proof-of-principle of a phononic SEIRA sensor, I believe the significance and impact of the manuscript justify its publication in Nature Communications.

Further optimization to enhance the detection sensitivity to approach the state-of-the-art MIR detectors is crucial for making it truly practical, though this apparently goes far beyond the scope of the current study. Therefore, I strongly recommend that the authors include a qualitative or semi-quantitative discussion on the upper limits of detection sensitivity of the phononic detector relative to MCTs in the first paragraph of the Discussion section (Page 11). This addition would provide valuable context for understanding the potential and limitations of the methodology for further follow-up studies.

Reviewer #2

(Remarks to the Author)

The authors have thoroughly revised their manuscript in response to the comments provided by previous referees. After reviewing the revised manuscript alongside the authors' detailed point-by-point responses to the referees, I recommend publication in Nature Communications. To reiterate my assessments:

Noteworthy results: As the authors noted in their response letter, unlike previous implementations (e.g. SRRs or other metallic elements) that relied on large plasmonic SEIRA detectors, this work successfully demonstrates the use of an ultra-compact single detector.

The manuscript is well-supported by rigorous and detailed theoretical analysis and numerical simulations that corroborate the experimental observations. The prediction of additional sensitivity improvements at subwavelength scales provides deeper scientific aspects to this research.

The proposed on-chip phononic SEIRA platform has implications for the development of compact, ultra-sensitive chemical sensor chips. Future integration with on-chip mid-IR light sources like quantum cascade lasers could enable on-site molecular and gas sensing applications.

Based on the very thorough revisions and the responses to the referees' comments, the manuscript fully addresses previous concerns and also provides strong scientific contributions to the field of SEIRA spectroscopy.

Reviewer #3

(Remarks to the Author)

I have read the detailed response to reviewers and the revised manuscript, and I am satisfied with most of the modifications done by the authors. I understand that some questions go beyond the scope of the paper and could not be done in a realistic timescale.

I have one major comment (but easily corrected) on the revised manuscript and sentences that were introduced. It is with regards to the comparison with far-field SEIRA "that uses LN-cooled MCT". It is an interesting point that the device of the authors work at room-temperature.

But it is an exaggeration to say that other far-field SEIRA performances are based on a cooled detector. Many published works would have the same conclusions with a room temperature DTGS detector, or room temperature microbolometers (see for instance Opt. Express 23, 5670-5680 (2015)). There are other criteria of comparison such as the SEIRA enhancement, the reflectivity difference, the reflectivity difference per nanometer that are independent of the choice of the detector, which plays a role on the detection threshold.

So I recommend to modify/suppress lines 296-297.

I recommend publication of the manuscript after considering this comment.

Reviewer #1 (Remarks to the Author):

We are grateful to the reviewer's recommendation in publishing the manuscript.

... and impact of the manuscript justify its publication in Nature Communications.

Further optimization to enhance the detection sensitivity to approach the state-of-the-art MIR detectors is crucial for making it truly practical, though this apparently goes far beyond the scope of the current study. Therefore, I strongly recommend that the authors include a qualitative or semi-quantitative discussion on the upper limits of detection sensitivity of the phononic detector relative to MCTs in the first paragraph of the Discussion section (Page 11). This addition would provide valuable context for understanding the potential and limitations of the methodology for further follow-up studies.

We address this comment in the revised manuscript together with the corrections made in response reviewer #3 (see below).

Reviewer #3 (Remarks to the Author):

We are grateful to the reviewer's recommendation in publishing the manuscript.

I have read the detailed response to reviewers and the revised manuscript, and I am satisfied with most of the modifications done by the authors. I understand that some questions go beyond the scope of the paper and could not be done in a realistic timescale.

I have one major comment (but easily corrected) on the revised manuscript and sentences that were introduced. It is with regards to the comparison with far-field SEIRA" that uses LN-cooled MCT". It is an interesting point that the device of the authors work at room-temperature. But it is an exaggeration to say that other far-field SEIRA performances are based on a cooled detector. Many published works would have the same conclusions with a room temperature DTGS detector, or room temperature microbolometers (see for instance Opt. Express 23, 5670-5680 (2015)). There are other criteria of comparison such as the SEIRA enhancement, the reflectivity difference, the reflectivity difference per nanometer that are independent of the choice of the detector, which plays a role on the detection threshold. So I recommend to modify/suppress lines 296-297. I recommend publication of the manuscript after considering this comment.

We address the comments of both reviewers with the following modified discussion paragraph:

“We note that the sensitivity of on-chip SEIRA detection has not yet reached the state-of-the-art far-field resonant SEIRA experiment **utilizing nitrogen-cooled mercury cadmium telluride (MCT) detector, where a sensitivity of about 500 molecules has been demonstrated³⁵. This can be attributed to two factors: the use of a non-resonant split-gate design and the mismatch between the HPhP resonance localized in the gap of split-gate and the molecular vibrational resonances. Future optimization of the device design for resonant detection could significantly boost sensitivity. This optimization includes adjusting the h-BN layer thicknesses, modifying bottom gate lengths, and fine-tuning the split-gate gap size. **Importantly, the room-temperature operation and the complementary metal-oxide-semiconductor (CMOS) compatibility of graphene detectors³⁶ represent progress toward compact on-chip sensors functioning under ambient conditions.**”**